# Driver Intent-Based Intersection Autonomous Driving Collision Avoidance Reinforcement Learning Algorithm

**DOI:** 10.3390/s22249943

**Published:** 2022-12-16

**Authors:** Ting Chen, Youjing Chen, Hao Li, Tao Gao, Huizhao Tu, Siyu Li

**Affiliations:** 1School of Information Engineering, Chang’an University, Xi’an 710064, China; 2Key Laboratory of Road and Traffic Engineering of the Ministry of Education, College of Transportation Engineering, Tongji University, Shanghai 201804, China

**Keywords:** self-driving vehicles, latent states, variational autoencoder, deep reinforcement learning

## Abstract

With the rapid development of artificial intelligent technology, the deep learning method is widely applied to predict human driving intentions due to its relative accuracy of prediction, which is one of critical links for security guarantee in the distributed, mixed driving scenario. In order to sense the intention of human-driven vehicles and reduce the self-driving collision avoidance rate, an improved intention prediction method for human-driving vehicles based on unsupervised, deep inverse reinforcement learning is proposed. Firstly, a contrast discriminator module was proposed to extract richer features. Then, the residual module was created to overcome the drawbacks of gradient disappearance and network degradation with the increase in network layers. Furthermore, the dropout layer was generated to prevent the over-fitting phenomenon in the whole training process of the GRU network, so as to improve the generalization ability of the network model. Finally, abundant experiments were conducted on datasets to evaluate our proposed method. The pass rate of self-driving vehicles with conservative driver probabilities of *p* = 0.25, *p* = 0.4, and *p* = 0.6 improved by a maximum of 8%, 10%, and 3%, compared with the classical method LSTM and VAE + RNN. It indicates that the prediction results of our proposed method fit more with the basic structure of the given traffic scenario in a long-term prediction range, which verifies the effectiveness of our proposed method.

## 1. Introduction

With the development of artificial intelligence and electric vehicles, in the process of converting from a manual driving transportation system to an autonomous driving transportation system, there will inevitably be a stage of mixed driving with different levels of intelligent carriers. The essence of mixed traffic flow is the coexistence of various driving behaviors, without standards to speak of, but the lack of uniformity will lead to difficulties in decision-making. Since human behavior is the most difficult to predict, autonomous vehicles usually adopt conservative driving strategies in mixed traffic flow scenarios, resulting in them constantly being overtaken by manual driving vehicles in the congested traffic flow, and the efficiency of passage is difficult to guarantee [1,2]. How to perceive the intentions of manual driving vehicles more intelligently and further improve the transportation efficiency of autonomous driving vehicles under the condition of ensuring safety has become a challenging task.

Taking the uncontrolled T-intersection scenario as an example, in the absence of critical guidance facilities such as traffic lights, self-driving vehicles must interact with other vehicles in the main lane if they want to merge safely and efficiently from the T-intersection to the main lane [3]. Since each traffic participant has its own driving strategy and driving style, autonomous vehicles need to perceive the hidden information of the surrounding vehicles and plan their own reasonable behavior trajectory accordingly [4,5]. If the latent traits of a human driver are known, that is, whether the driver is aggressive or conservative, the self-driving vehicles adopt different driving strategies correspondingly, so as to obtain a better balance of safety and efficiency. Therefore, it is crucial to accurately judge the driving traits of human-driven vehicles.

The latent traits of human-driven cars are divided into intent estimation and trait estimation [6]. Intent estimation typically uses methods such as probabilistic graphical models and non-parametric belief trackers to predict the future actions of other drivers, thereby providing information for the next trajectory planning for self-driving cars. Song W et al. [7] proposed the use of the continuous hidden Markov model to predict both the high-level motion intentions (e.g., turn right, turn left) and the low-level interaction intentions (e.g., the yield status of related vehicles). Dong C et al. [8] utilized the approach based on the probabilistic graphical model (PGM) to efficiently estimate the intent of self-driving cars and interact with them in ramp merging scenarios, even without communication between vehicles. Bai H et al. [9] proposed an online planning method to estimate latent pedestrian intentions using a partially observable Markov decision process (POMDP) for self-driving vehicles to make the systematical and robust decisions in the presence of many pedestrians.

Trait estimation infers the driving characteristics of drivers, such as driving style, driving preference, fatigue status, and degree of distraction, etc. Supervised and unsupervised learning are usually utilized to classify driving characteristics. Morton et al. [10] learned the latent traits of driver characteristics and input the traits and current environmental states into the policy network to produce multi-modal behaviors. However, the input of the strategy network represents the short-term state of the current vehicle, which does not adequately represent the long-term nature of these strategies. Ma et al. [11] utilized supervised learning to classify the traits of human-driven vehicles for autonomous vehicle navigation at intersections, but trait labels are expensive to obtain and do not typically exist in most real driving datasets. Second, the driving policies are trained with ground truth trait labels rather than predictive features. When feature classifiers and policy networks are combined, generating errors in testing and cascading leads to severe performance degradation. Another class of characteristic driving feature learning methods is variational auto-encoding (VAE) and its variants [12,13]. Moreover, conditional VAE (CVAE) is widely used in trajectory predictions of the pedestrian and vehicle trajectory prediction because discrete potential states represent different behavioral patterns, such as braking and turning. Salzmann T et al. [14] proposed a generative multi-intelligent trails prediction method that generated a probability distribution for the agent’s motion planning and decision-making. Ivanovic B et al. [15] proposed CVAE to predict human behavior, which generates multi-modal probability distributions on future human trails based on past human–robot interactions and the future actions of candidate robots. Feng X et al. [16] proposed a model that estimates potential driver characteristics and generates a CVAE for multi-modal trail prediction. These behavior patterns change frequently, and the driving characteristics of each driver are persistent. Bowman et al. [17] introduced a recurrent neural network-based VAE model to simulate the latent properties of sentences and explicitly modeled the overall properties of sentences. Liu et al. [18] was inspired by learning the traits of drivers from trails, encoding the trails of drivers as driving features of drivers using a proposed RNN-based VAE. However, the VAE network has a limited ability to characterize the approximate posterior distribution, resulting in low quality of the generative latent variables. These drawbacks largely limit the ability of the latent variables in VAE to express serial information, which is unique to the vehicle trails.

In recent years, contrast learning has been widely applied to learn feature information from continuous data such as video and pedestrian trajectories [19]. Wang X et al. [20] proposed a Siamese triplet network with rank loss function to train the visual representation method of the convolutional neural network (CNN). Liu Y et al. [21] introduced a social contrastive loss that regularizes the extracted motion representation by discerning the ground truth positive events from synthetic negative ones. Zhe Xie et al. [22] introduced contrast learning into the VAE model and utilized contrast loss to improve the ability of the VAE model to represent features and learn the unique features of different users.

Inspired by contrast learning, this paper improves the VAE + RNN model and introduces contrast loss and the residual network to form the contrastive-ResNet-VAE model (C-ResNet-VAE) so that autonomous vehicles can better avoid people when driving through the uncontrolled intersections. The optimized contrast loss not only enhances the model’s ability to separate different features but also improves the model’s ability to learn the potential features of different drivers from the trajectory. Our main contributions are as follows:
(1)In order to improve the ability to learn the potential features of different drivers from the trajectory, contrast learning was proposed into the model, which used the minimization of contrast loss to learn the exclusive features of different drivers in the driver trajectory and enhanced the ability of the model to separate different features.(2)We introduced residual modules in the GRU model to capture detailed feature information with strong representational power. These stacked residual units greatly improved the training efficiency, ensuring that the network in the latter layer captured more feature information than the previous layer, reducing information loss. Moreover, the dropout layer was introduced to prevent the over-fitting phenomenon in the whole training process of the gated recurrent unit (GRU) network to improve the generalization ability of the network model.

## 2. Preliminaries

Kingma et al. [12] proposed VAE as a deep generative model in 2013. The VAE model contains two parts: the encoder and decoder. The encoder makes variational inferences with the input and generates an approximate posterior probability distribution of hidden variables. The role of the decoder is to recover the hidden variables to an approximate probability distribution of the input. The overall framework of the VAE model is shown in Figure 1.

x is the real sample, z is the hidden variable, and $\hat{x}$ denotes the output of the decoder. Moreover, φ represents the parameters in the encoder, and θ represents the parameters in the decoder. Furthermore, μ and $\sigma^{2}$ represent the mean and variance of the approximate posterior probability distribution of z. Specifically,x is an observable random vector of the high-dimensional space, and z is an unobservable random vector of relatively low-dimensional space. The high-dimensional represents the observable x-space in the VAE model. The low-dimensional representation decoder reduces the dimensionality of the hidden variable z. The VAE model sample generation is divided into two processes: the encoder infers the approximate distribution process $q_{\varphi }(z|x)$ of the hidden variables, and then the decoder restores the hidden variables z to a process $P_{\theta }(x|z)P_{\theta }(z)$ similar to the probability distribution of the input sample. $q_{\varphi }(z|x)$ denotes the approximate posterior probability distribution of z.

Due to the fact that the encoder is unable to obtain the prior probability distribution of the hidden variable z, the VAE model introduces a learning model $q_{\varphi }(z|x)$ instead of the true posterior distribution of the hidden variable z; it assumes $q_{\varphi }(z|x)$ obeys the ordinary normal distribution. Meanwhile, for calculation convenience, assume that the implicit variable prior distribution $P_{\theta }(z)$ follows the standard normal distribution. The optimization goal of the encoder is to get $q_{\varphi }(z|x)$ as close as possible to $p_{\theta }(x|z)$. $p_{\theta }(x|z)$ denotes the approximate posterior probability distribution of x.

The VAE model adopts the Kullback–Leibler (KL) divergence [23] to evaluate the similarities between them. Thus, the encoder optimization objective is expressed as:(1)$$\arg\min D_{KL}(q_{\varphi }(z|x)∥p_{\theta }(x|z))=\log(P_{\theta }(X))-L(\theta ,\varphi ;X)$$
where L(θ,φ;X) is the variational lower bound function of the VAE model, $\log(P_{\theta }(X))$ is the constant of the encoder, and $P_{\theta }(z)$ denotes the hidden variable prior probability distribution.

The VAE model adopts the encoder to learn the posterior distributed parameter mean μ and variance $\sigma^{2}$ of the latent variables from the input sample, and then performs sampling to obtain the latent variables from the distribution. Since the sampling operations are irreducible, the reparameterization technique was proposed in literature [24]. Specifically, the process of sampling from a normal distribution $z\sim N(\mu ,\sigma^{2})$ is replaced with ε acquisition from a standard normal distribution, and the parameter transformation z=μ+ε×σ is utilized to obtain the latent variables. ε denotes standard normal distribution. After reparameterization transformation, the sampling process is accessible, and the model is able to be trained.

## 3. Proposed Methods 

We mostly researched two-way two lanes at an uncontrollable T-Intersection, shown in Figure 2. The vehicle in the lower lane turned left, whereas, the vehicle in the higher lane turned right. An autonomous vehicle turned to the higher lane in a safe way to turn right. More specifically, the blue vehicle was conservative; the red vehicle was aggressive; and the yellow vehicle was autonomous. The conservative vehicle gave way to the autonomous vehicle, but the aggressive vehicle ignored the autonomous vehicle and continued forward.

Liu et al. adopted VAE + RNN to infer drivers’ potential features from the trail of vehicles. However, the VAE network is restricted to deducing the abilities of potential features. We introduced contrastive learning and residual modules based on the VAE + RNN network. Meanwhile, we leveraged C-ResNet-VAE + RNN networks to extract the potential states of various drivers from original driving trails and clusters in an unsupervised way. Figure 3 shows the whole framework of the network. Both datasets and unsorted potential states were simultaneously inputted to the contrastive learning classifier, and contrastive losses were calculated. The contrastive learning classifier optimized encoded parameters in a back propagation way. Then, potential features with the learned features of drivers and all the states of the vehicles to a navigation strategy were submitted. The strategy network contained the GRU network with an attentive module, trained by model-free reinforcement learning. According to inferred features, autonomous vehicles adjusted strategies when interacting with a variety of drivers to perform efficiently.

### 3.1. The Network of Potential Features with an Unsupervised Cluster Module

In order to obtain the potential features of every driver from the trails of drivers, having the C-ResNet-VAE + GRU network extract the styles of driving was proposed. The C-ResNet-VAE network is composed of an encoder, contrastive learning classification, and decoder. The encoder squeezes the collected trail x and forms the distribution of the potential variable z. The decoder rebuilds the trails from potential features. The potential variable z is unsorted to get the potential states $\tilde{z}$. We inputted both the positive and negative samples, (x,z) and $\left(x,\tilde{z}\right)$, respectively, to the classifier. Moreover, we calculated contrastive losses and optimized encoded parameters in a back propagation way. The potential features with an unsupervised cluster module are shown in Figure 4.

#### 3.1.1. Contrast Learning

The data of drivers’ trails contains generous similar samples, so the VAE module poorly analyzes the drivers’ particular features. If it fails to distinguish the different features of every driver, it will influence the results of autonomous driving in a decision network, and the decoder will rebuild the original inputs from the VAE module. If the exclusive target of the module is to reconstruct sequences, something necessary and remarkable, such as drivers’ personal information, will be neglected. Here, we aimed to use contrastive learning to train the VAE module, which enhances the representational ability of potential variables.

We compared the contrastive losses, which the potential variable learns in different drivers by using the contrastive classifier $G_{\omega }$, and gained the effective and crucial features of $x_{u}$. We defined a set of coordinates $\left(x_{u},z_{u}\right)$ as positively matching; that is to say, the trail $x_{u}$ generated the corresponding potential encoded variable $z_{u}$. 

The formulation of contrastive losses is as follows:(2)$$L_{\omega ,\phi }(x_{u},z_{u})=-\sum_{t=1}^{T_{u}}\left[E_{z_{u}\sim Q_{\phi }(z_{u}|x_{u})}\log(\sigma (G_{\omega }{\left(x_{u},z_{u}\right)}^{\left(t\right)}))\right]+E_{{\tilde{z}}_{u^{\prime }}\sim Q_{\phi }({\tilde{z}}_{u}|x_{u^{\prime }})}\log(\sigma (1-G_{\omega }{\left(x_{u^{\prime }},{\tilde{z}}_{u}\right)}^{\left(t\right)}))$$
where u defines the set of all driver trajectories, E denotes the evidence lower bound, t denotes the timestep, and $T_{u}$ denotes the number of items in the driver trajectories.

When the contrastive loss is minimum, i.e., $L_{\omega ,\phi }(x_{u},z_{u})$, the classifier $G_{\omega }$ distinguishes positive matching with negative matching efficiently. The potential variables that decoders infer will explicitly collect more significant individual information.

#### 3.1.2. Feature Extraction Based on Residual

We improved the network based on VAE + GRU and presented a feature extraction network based on residual structure to capture detailed feature information with strong representation ability. Generally speaking, researchers mostly increase the number of network layers to improve the richness of feature information; with more layers or a wider network, the abstract level of feature information will gradually increase. From the initial acquired edge, information gradually becomes more representational semantic information. However, the number of network layers are not deepened infinitely. In the process of deepening the network layers, the model will have a foremost outcome. If the number of network layers continues to increase, the loss will increase accordingly. He et al. [25] presented the ResNet model with the basic idea that residual mapping is easy to optimize, so the ResNet model skips over the convolutional layers and forms the residual unit by using rapid connections. These stacked residual units greatly improve the training efficiency, ensuring that the next layer obtains more feature information than the previous layer and solves the degradation problem caused by the deepening of the network in a large part. The residual unit consists of two main branches. The first branch is identity mapping and the other is residual learning. If the input value of the residual unit is *x*, the feature mark obtained by residual learning is *F*(*x*), and the output value is *H*(*x*), then this unit is expressed as:(3)H(x)=x+F(x)

The structure of residual unit is shown in Figure 5.

### 3.2. Intensive Learning Decision-Making Module Network

As shown in Figure 6, the decision-making module is a GRU network with an attention module. The vehicle status includes the potential characteristics of the driver, all observable vehicle site coordinates, and the site and speed of the autonomous vehicle. Here is how it works. First of all, the vehicle status is inputted to the attention module, which distributes the attention weight to every surrounding vehicle. Then, the weight characteristics counted by each vehicle and the site and speed of the autonomous vehicle are fed into the GRU network. Finally, the hiding states in the GRU network are put into a full connected layer to obtain the value function and the policy function. In this paper, we used a policy gradient method of model-free intensive learning, which we called the proximal policy optimization (PPO) algorithm, to learn the value function and policy function [26] and utilized the approach in this literature [27] to achieve the PPO algorithm.

#### Proximal Policy Optimization (PPO) Algorithm

This method alternates between the sample data by interacting with the environment and using random gradient rise to optimize the “agent” objective function. Assume $r_{t}(\theta )=\frac{\pi_{\theta }(a_{t}|s_{t})}{\pi_{\theta old}(a_{t}|s_{t})}$, where $a_{t}$ is the current action, and $s_{t}$ is the current state. 

The objective function of the PPO algorithm is:(4)$$L(\theta )=\hat{E}t[\frac{\pi_{\theta }(a_{t}|s_{t})}{\pi_{\theta old}(a_{t}|s_{t})}{\hat{A}}_{t}]=\hat{E}[r_{t}(\theta ){\hat{A}}_{t}]$$

The ${\hat{A}}_{t}$ is an estimated value of the preponderance function of time step(*t*) of $\left(s_{t},a_{t}\right)$, and $\hat{E}t$[…] represents the expected value for a batch of samples. The policy gradient algorithm is more efficient in continuous action spaces [28].

## 4. Experiments and Results 

Our simulation environment was at the uncontrolled T-intersections, and we assumed that there were *n* cars moving toward opposite directions in a two-way street and that all of the vehicles that were controlled by the Intelligent Driver Model (IDM) [29] never took turns or changed lanes. The drivers with a conservative driving style varied their front gaps from the preceding vehicles between 0.5 m and 0.7 m and had the desired speed of 2.4 m/s. The drivers with an aggressive driving style varied their front gaps between 0.3 m and 0.5 m and had the desired speed of 3 m/s. The ego car started at the bottom of the T-intersection and then took a right turn to merge into the upper lane without colliding with other cars. If the ego car encountered the other cars, the conservative drivers would yield to the ego car, while the aggressive drivers would ignore and collide with the ego car. The ego car with a fixed right-turn path was controlled by a longitudinal proportional–derivative (PD) controller, whose desired speed was set by the policy network. 

Let the state of the ego vehicle successfully making a full right turn be $S_{success}$ and the vehicle successfully making a full right-turn have a small reward on the speed, where $r_{speed}(s)=0.05\times ∥v_{auto}{∥}_{2}$; $v_{auto}$ means the speed of self-driving vehicles. Meanwhile, let the state of the ego vehicle colliding with other vehicles be $S_{fail}$ and let the vehicle have a constant penalty on the speed, where $r_{step}=-0.0013$. Otherwise, we set the length of the cars as 5 cm and the width of the cars as 2 cm. This is to encourage the ego car to reach the goal of making a full right turn as soon as possible. The reward function is defined as:(5)$$r(s,a)=\{\begin{array}{c} 2.5,\\ -2,\\ r_{speed}(s)+r_{step}, \end{array}\begin{array}{c} s\in S_{success}\\ s\in S_{fail}\\ others \end{array}$$

### 4.1. Datasets

The dataset used in this paper is a randomly generated trajectory dataset in Python produced by Liu et al. Because using Python to generate the dataset not only allows one to set the type of dataset needed, such as the number of entries allocated to radical and conservative trajectories in the program, it also saves the cost of obtaining these datasets. It contains approximately 700,000 driving trajectories from two types of drivers. We set the train/test split ratio is 2:1. We trained the policy model with 466,667 random trajectory data and tested with the other 233,333 trajectory data. We set the decaying learning rate to 5 × 10^−4^ and the weight of the KL divergence loss to β 5 × 10^−8^. 

### 4.2. Unsupervised Clustering Representations of Latent Driver Traits

In this paper we proposed a network of C-ResNet-VAE + GRU and compared it with the VAE + GRU network [18] and GRU network [11]. Both the study [18] and we utilized a GRU as the encoder and GRU the as decoder, while [10] utilized GRU as the encoder and the multilayer perceptron (MLP) as the decoder. In addition, in order to verify the effectiveness of the residual module and reinforcement learning, we trained the ablation experiments without the residual module and comparative loss.

We trained two methods for 500 epochs and then utilized a set of test trajectories as inputs to act as encoders to measure if the latent trait effect was good or bad. The unsupervised classifier results of both methods are shown in Figure 7.

The red areas represent aggressive drivers, the blue areas represent conservative drivers, and the middle, which is not fully separated out, usually contains very short trajectories and trajectories with vague front clearance. The lines closer to the boundary represent vehicle trajectories where the car has moved out of the range and we cannot see. As shown in Figure 7a–c, the unsupervised classifier we proposed successfully classified most of the result differences in driving styles. From Figure 7b, it can be seen that there are still a large number of blue areas in the red region that have not been successfully separated out, and from Figure 7c, it can be seen that there are significantly fewer blue areas in the red region. It can be seen that the method in this paper can better separate the shorter trajectories and the fuzzy front gap trajectories and is more capable of separating the two different characteristics of conservative and aggressive driver styles. It has a better effect, compared with the methods [10,18] proposed and has the ability to separate the traits of aggressive and conservative driving styles. Study [10] utilized GRU as the encoder to obtain the two latent vectors. We assumed that the vehicles only considered current states and actions and the encoder only considered the short-term information of the vehicle acceleration, such as the latent traits of drivers.

Therefore, trajectories with different potential characteristics were mostly gathered together and not classified successfully. Study [18] utilized the VAE + RNN network, where VAE had limited approximate posterior distribution trait abilities. Hence, there were some poor sample qualities generated. The C-ResNet-VAE network we proposed improved the approximate posterior distribution trait abilities of the VAE network, obtained richer information from vehicle trajectory, was simpler and better suited for the unsupervised classifier, as well as distinguished the difference between different trajectories, so it had a better performance.

From the results in Figure 7c–e, it can be seen that there are significantly more blue blocks in the red area without the addition of the contrast discriminator module and the residual module, and there are still many driving trajectories with different potential features fused together. With the addition of the contrast discriminator module and the residual module, the separation ability is enhanced, and the clustering effect is better. Thus, the effectiveness of the method is verified.

### 4.3. Decision Results of Self-Driving Strategies 

We used two baselines as comparative experiments:
(1)The supervised learning with labels proposed by Ma X et al. [11], which trained a supervised trait predictor and a reinforcement learning policy with truth trait labels separately and combined them at test time. (2)The strategy of [10,18], and our model all utilize unsupervised methods to infer the potential state of drivers to make reinforcement learning decisions. 

In addition, we used a reinforcement learning policy directly trained with truth labels as a baseline. We ran experiments with different proportions of two types of drivers and tested four models with 500 random cases. The percentage of auto-vehicles successfully taking a right turn to merge into the upper lane, colliding with surrounding vehicles and completing overtime, were calculated, respectively, where overtime refers to a situation where the car failed to make a right turn within the allotted time and did not crash. The results are shown in Table 1, Table 2 and Table 3.

*p* is the probability for each surrounding driver to be conservative. The task difficulty decreased as *p* increased. From the experimental results, the collision rate of the proposed potential state feature extraction method was lower than the other three methods, and the successful completion of the task accounted for a higher proportion, which is closer to decision-making under the real label training. The main reason was that our method effectively extracted the trait differences of the surrounding drivers, which made better use of the reinforcement learning for the decision-making of the ego vehicle. Our policy was able to utilize the existing trait representation and focused more on the decision-making of the ego vehicle, which led to better navigation performance. The model in [10] had a low success rate when *p* value became smaller. The reason is that the latent representation did not distinguish between different traits and only provided very limited useful information to learners. For Ma et al. [11], both strategies had good performance when tested separately. However, when the two modules were combined together, intermediate and cascading errors significantly lowered the success rates. Since the policy was trained with true traits, it failed easily whenever the trait classifier made a small mistake. The model in Liu S et al. [18] had limited representation ability, low separation ability for some fuzzy trajectories (very short trajectories and the trajectories with fuzzy front gaps), and could not learn unique traits of different drivers well. The simulation process of the self-driving car successfully making a full right-turn on the upper lane at the uncontrolled T-intersection is shown in Figure 8. The cars in the two lanes went in opposite directions; the conservative cars are in green, and the aggressive cars are in red. 

At time t, it can be seen from Figure 8a that when the self-driving car met an aggressive red vehicle, it gave way to it. When it met a green conservative vehicle, it passed it, and finally, the self-driving car successfully merged into the car lane and completed the right turn.

At time *t* + 1, it can be seen from Figure 8b that the autonomous vehicle successfully identified the red aggressive vehicle when passing the drop-off lane and passed the green conservative vehicle when it encountered it. The unsuccessful recognition of the red aggressive vehicle at the entry lane resulted in a collision.

At time *t* + *n*, it can be seen from Figure 8c that the autonomous vehicle recognized the red aggressive vehicle when passing the drop-off lane and passed the green conservative vehicle when it encountered it; however, failed to make a judgment when it was about to enter the drop-off lane and failed to take action, resulting in a timeout. Timeout *n* was set to 50 s.

## 5. Conclusions

In this paper, the C-ResNet-VAE network was proposed to improve the existing deficiency of the VAE + RNN model and learn the potential characteristics of drivers from vehicle trajectories. Then, the potential characteristics and vehicle status were used to learn the trajectory prediction of autonomous vehicles at uncontrolled T-intersections. The introduction of contrast loss better learned the exclusive characteristics of the drivers from the vehicle trajectory and ensured the personalized and distinctive characteristics of the drivers; the residual network was added to the latent variable of feature extraction to improve the ability of feature extraction and prevent the gradient from disappearing. Experiments showed that the proposed method better separated the potential characteristics of drivers with different styles, received more exclusive characteristics of drivers from the trajectory, and improved the collision probability at uncontrolled intersections. However, the fuzzy driving trajectory was not successfully distinguished, and lane change and turning were not studied, which is the direction of our future research.

## Figures and Tables

**Figure 1 sensors-22-09943-f001:**
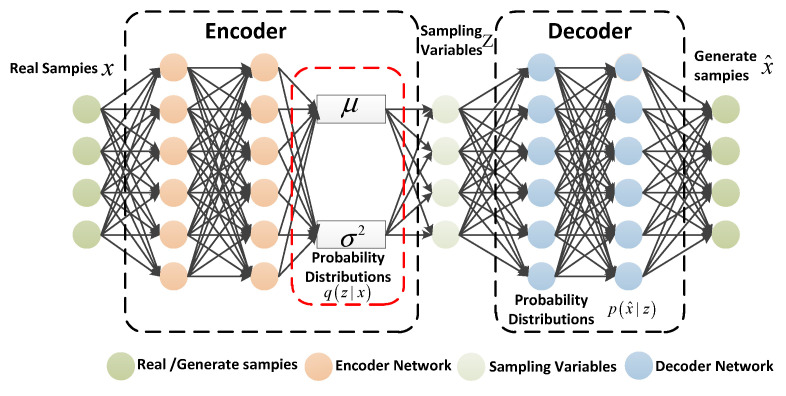
The overall framework of the VAE model.

**Figure 2 sensors-22-09943-f002:**
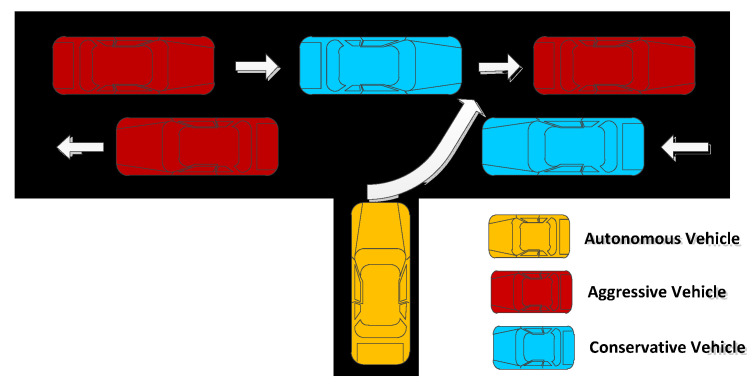
An uncontrollable T-intersection.

**Figure 3 sensors-22-09943-f003:**
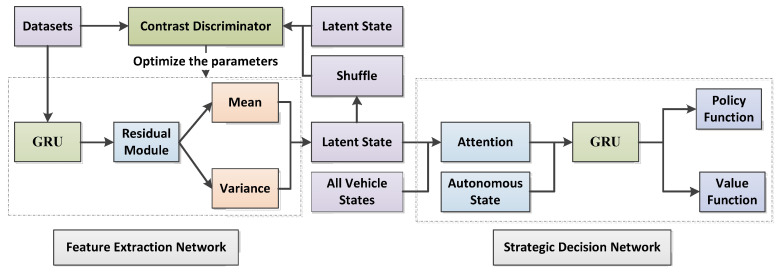
The whole framework of the C-ResNet-VAE + RNN network.

**Figure 4 sensors-22-09943-f004:**
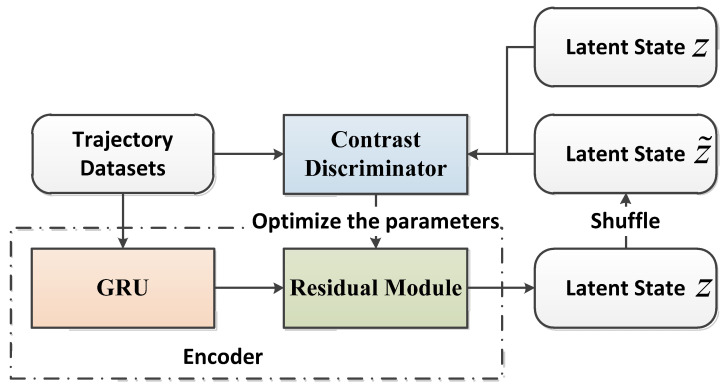
The potential features with an unsupervised cluster module.

**Figure 5 sensors-22-09943-f005:**
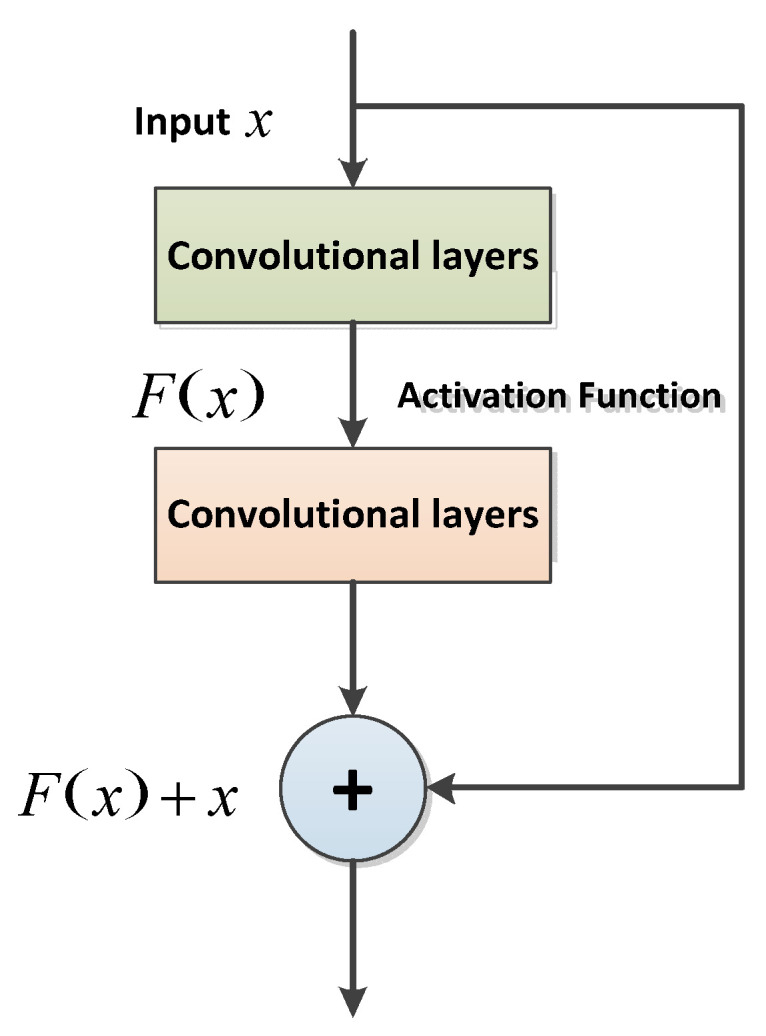
The structure of the residual unit.

**Figure 6 sensors-22-09943-f006:**
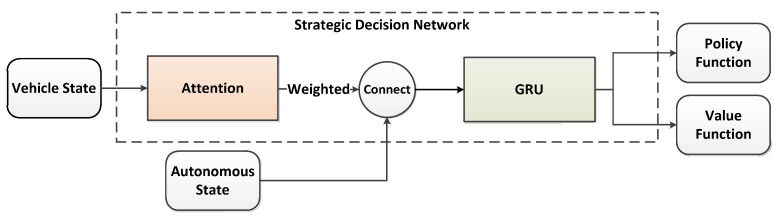
Strategy decision-making module.

**Figure 7 sensors-22-09943-f007:**
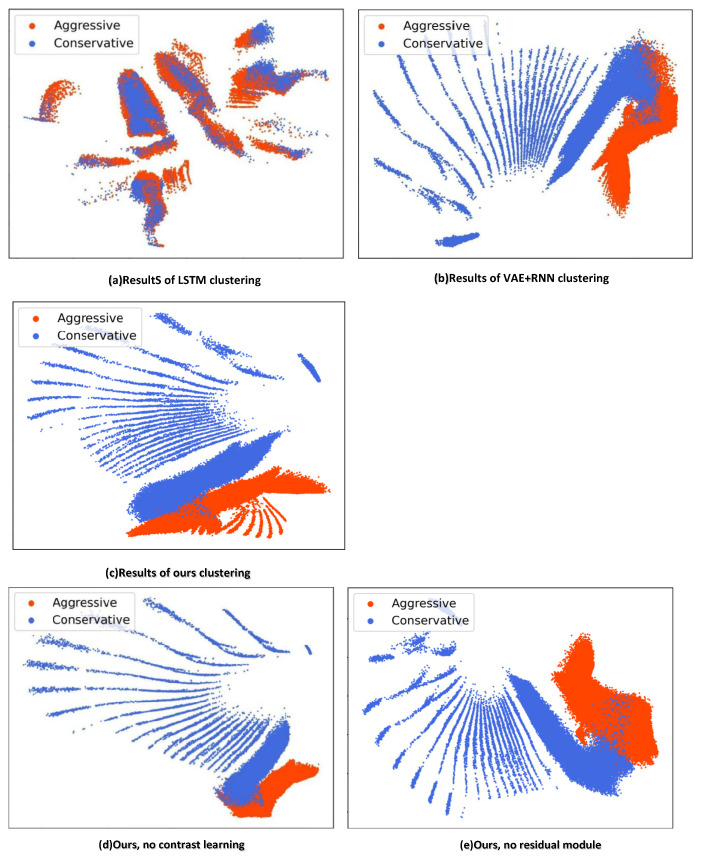
Comparison of unsupervised clustering result. The *x* and *y*-axes are the horizontal and longitudinal displacements in meters.

**Figure 8 sensors-22-09943-f008:**
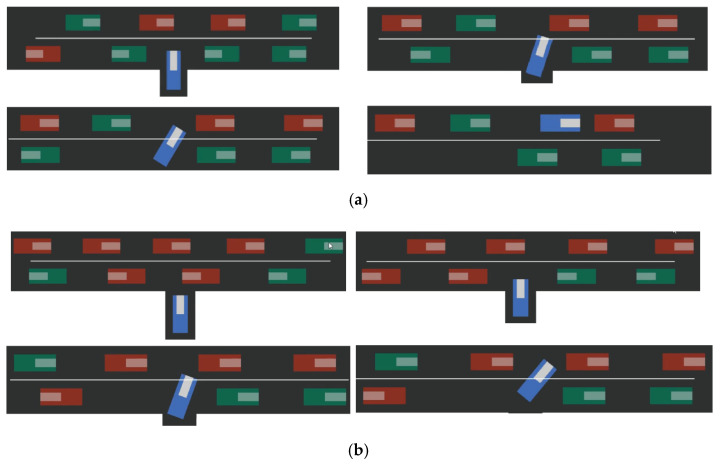
Process of the self-driving car making a full right-turn on the upper lane. (**a**) self-driving car successfully makes a full right-turn on the upper lane at time *t*. (**b**) Self-driving car collide at time t+1. (**c**) Self-driving car overtime at time t+n.

**Table 1 sensors-22-09943-t001:** Objective evaluation decision results of self-driving strategies with aggressive drivers *p* = 0.25.

Models	Success (%)	Timeout (%)	Collision (%)
True Labels	85	1	14
GNN	66	18	16
LSTM	70	16	14
VAE + RNN	73	15	12
Ours	78	10	12

**Table 2 sensors-22-09943-t002:** Objective evaluation decision results of self-driving strategies with aggressive drivers *p* = 0.4.

Models	Success (%)	Timeout (%)	Collision (%)
True Labels	91	4	5
GNN	73	22	5
LSTM	74	20	6
VAE + RNN	80	12	8
Ours	84	11	5

**Table 3 sensors-22-09943-t003:** Objective evaluation decision results of self-driving strategies with aggressive drivers *p* = 0.6.

Models	Success (%)	Timeout (%)	Collision (%)
True Labels	97	1	2
GNN	87	10	3
LSTM	95	3	2
VAE + RNN	96	2	2
Ours	98	0	2

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
