# Peer review of "Driver Intent-Based Intersection Autonomous Driving Collision Avoidance Reinforcement Learning Algorithm"

_sensors, 2022, doi:10.3390/s22249943_

Round 1

Reviewer 1 Report

This paper proposed an intention prediction method for human-driving vehicles based on unsupervised deep inverse reinforcement learning. First, a contrast discriminator module for extracting features is proposed into the feature extraction network. Then, the residual module is introduced to overcome the drawbacks of gradient disappearance and network degradation with the increase of network layers. Furthermore, the Dropout layer is generated to prevent the over-fitting phenomenon in the whole training process of GRU network. Finally, simulation experiment is conducted to evaluate the proposed method. The work is interesting. However, the results are limited, and the English of the article should be improved. According to the results of the paper, the advantages of the proposed method are not particularly outstanding.

Author Response

Thank you for your letter and for the reviewers’ comments concerning our manuscript entitled “Driver Intent Based Intersection Autonomous Driving Collision Avoidance Reinforcement Learning Algorithm”. Those comments are all valuable and very helpful for revising and improving our paper, as well as the important guiding significance to our researches. We have studied comments carefully and have made correction which we hope meet with approval. The main corrections in the paper and the responds to the reviewer’s comments are as flowing:

Responds to the reviewer’s comments:

Reviewer #1

Comments and Suggestions for Authors

This paper proposed an intention prediction method for human-driving vehicles based on unsupervised deep inverse reinforcement learning. First, a contrast discriminator module for extracting features is proposed into the feature extraction network. Then, the residual module is introduced to overcome the drawbacks of gradient disappearance and network degradation with the increase of network layers. Furthermore, the Dropout layer is generated to prevent the over-fitting phenomenon in the whole training process of GRU network. Finally, simulation experiment is conducted to evaluate the proposed method. The work is interesting. However, the results are limited, and the English of the article should be improved. According to the results of the paper, the advantages of the proposed method are not particularly outstanding.

  1. Response to comment:

However, the results are limited, and the English of the article should be improved. According to the results of the paper, the advantages of the proposed method are not particularly outstanding.

 Response:

Special thanks to you for your good comments. We have properly revised the English of the article.

Main contributions are summarized as follows:

1) In order to improve the ability of learning the potential features of different drivers from the trajectory, a contrast learning model is presented, which utilized the minimization of contrast loss to learn the exclusive features of different drivers in the driver trajectory, and enhances the ability of the model to separate different features.

2) We proposed stacked residual units to capture detailed feature information with strong representational power, which greatly improve the training efficiency, ensure more rich feature information in latter layer than the previous layer, and reduce information loss.

3)The Dropout layer is created to prevent the over-fitting phenomenon in the whole training process of Gate Recurrent Unit(GRU) network, so as to improve the generalization ability of the network model.

Thank you very much for all your help and looking forward to hearing from you soon.

Reviewer 2 Report

In this manuscript, the authors present the study of using the C-ResNet-VAE network to predict the behavior of autonomous vehicles at an uncontrolled T-intersection. However, many of the descriptions and figures need further clarification. 

Please respond to all the line-by-line comments below.

Line 14, remove "the".

Line 22, define "P" here, otherwise, it makes no sense to the readers

Line 23, why do exclusively mention LSTM, but not another method that is listed in your tables?

Line 33, ",without" -> ", without".

Line 37, add space between "guaranteed" and "[1]", and the same for line 44's [3], line 47's [4], line 53's [6].

Line 44, please fix the pronoun "his".

Line 57, "motion_intentions" -> "motion intentions".

Line 71, "vehicle,which" -> "vehicle, which".

Line 89, define RNN.

Line 100, define CNN.

Line 106, define C-ResNet-VAE.

Line 118, define GRU.

Line 121, there's only one subsection in this section, you might just rename the section and remove the subsection.

Figure 1, please explain what different colors mean.

Lines 130-138, need clarification. Please use the same notations in the text as in Figure 1, change "x", "z" to "X", "Z", or the other way around. Define "q_{\phi} (z|x)". Define "P_{\theta}(x|z)" (make it consistent with what's in Figure 1). Define  "P_{\theta}(z)".

Line 131, "represent" -> "represents".

Line 134-135, please elaborate on low- and high-dimensional.

Line 145, define KL.

Equation 1 is incomplete.

Line 152, "obtains" -> "obtain"

Line 155, "noise \epsilon", what does it mean?

Line 188-191, please make sure the "x" and "z" are in line with the text.

Line 197, "samples.," -> "samples,".

Line 198, "analyses" -> "analyzes".

Line 206-208, please make sure "G", "x" and "z" are in line with the text, the same for line 212.

Equation 2, please define "u", "E", "t", "T_u".

Line 219, "the more" -> "more".

Line 228, "train" -> "training". "insure" -> "ensure".

Line 247, define PPO here instead of at line 254.

Line 253-255 is a repeat of what has just been said.

Equation 4, please define every single symbol that has not been defined.

Line 265, define IDM.

Line 267-269, is there a reason why such slow speeds are chosen for the experiments? 2.4 m/s is no faster than a jogger. Also, 0.3 -0.7 m is not a realistic range in real life. What is the length of the cars?

Line 273, define PD.

Line 277, define "v_auto".

Line 284, please provide more details about the datasets, "generated by Python" provides zero information.

Line 293, define MLP.

Figure 7 and lines 303-321, please describe in more detail what is shown in the figures. What do the lines in them mean? What are the x and y-axis? How do you quantify the difference between b, c, d, and e? Define "LSTM".

Line 333, define "overtime".

Table 3, please make sure the whole table is on a single page.

Line 358, define "fuzzy trajectories".

Line 360, there's no figure 9.

Line 361, the conservative cars are green, not blue.

Figure  8, please make there is no text in between the figures. Please explain what the four small figures mean in (a) and (b), and what the six figures mean in (c). 

Line 373, "t+n" what is the size of "n" here?

Line 387, "drivers;The" -> "drivers. The".

Line 387, "ResNet18", what does it mean?

Line 390, "and better gets more", please fix the grammar.

References:

Please carefully revise *ALL* the references- remove unnecessary symbols, such as [J], [C],"//", etc. Please add space or "." where it's needed. Please make sure the arXiv numbers are correctly shown.

Author Response

Thank you for your letter and for the reviewers’ comments concerning our manuscript entitled “Driver Intent Based Intersection Autonomous Driving Collision Avoidance Reinforcement Learning Algorithm”. Those comments are all valuable and very helpful for revising and improving our paper, as well as the important guiding significance to our researches. We have studied comments carefully and have made correction which we hope meet with approval. The main corrections in the paper and the responds to the reviewer’s comments are as flowing:

Responds to the reviewer’s comments:

Reviewer #2

Please respond to all the line-by-line comments below.

Line 14, remove "the".

Line 22, define "P" here, otherwise, it makes no sense to the readers

Line 23, why do exclusively mention LSTM, but not another method that is listed in your tables?

Line 33, ",without" -> ", without".

Line 37, add space between "guaranteed" and "[1]", and the same for line 44's [3], line 47's [4], line 53's [6].

Line 44, please fix the pronoun "his".

Line 57, "motion_intentions" -> "motion intentions".

Line 71, "vehicle,which" -> "vehicle, which".

Line 89, define RNN.

Line 100, define CNN.

Line 106, define C-ResNet-VAE.

Line 118, define GRU.

Line 121, there's only one subsection in this section, you might just rename the section and remove the subsection.

Figure 1, please explain what different colors mean.

Lines 130-138, need clarification. Please use the same notations in the text as in Figure 1, change "x", "z" to "X", "Z", or the other way around. Define "q_{\phi} (z|x)". Define "P_{\theta}(x|z)" (make it consistent with what's in Figure 1). Define  "P_{\theta}(z)".

Line 131, "represent" -> "represents".

Line 134-135, please elaborate on low- and high-dimensional.

Line 145, define KL.

Equation 1 is incomplete.

Line 152, "obtains" -> "obtain"

Line 155, "noise \epsilon", what does it mean?

Line 188-191, please make sure the "x" and "z" are in line with the text.

Line 197, "samples.," -> "samples,".

Line 198, "analyses" -> "analyzes".

Line 206-208, please make sure "G", "x" and "z" are in line with the text, the same for line 212.

Equation 2, please define "u", "E", "t", "T_u".

Line 219, "the more" -> "more".

Line 228, "train" -> "training". "insure" -> "ensure".

Line 247, define PPO here instead of at line 254.

Line 253-255 is a repeat of what has just been said.

Equation 4, please define every single symbol that has not been defined.

Line 265, define IDM.

Line 267-269, is there a reason why such slow speeds are chosen for the experiments? 2.4 m/s is no faster than a jogger. Also, 0.3 -0.7 m is not a realistic range in real life. What is the length of the cars?

Line 273, define PD.

Line 277, define "v_auto".

Line 284, please provide more details about the datasets, "generated by Python" provides zero information.

Line 293, define MLP.

Figure 7 and lines 303-321, please describe in more detail what is shown in the figures. What do the lines in them mean? What are the x and y-axis? How do you quantify the difference between b, c, d, and e? Define "LSTM".

Line 333, define "overtime".

Table 3, please make sure the whole table is on a single page.

Line 358, define "fuzzy trajectories".

Line 360, there's no figure 9.

Line 361, the conservative cars are green, not blue.

Figure  8, please make there is no text in between the figures. Please explain what the four small figures mean in (a) and (b), and what the six figures mean in (c).

Line 373, "t+n" what is the size of "n" here?

Line 387, "drivers;The" -> "drivers. The".

Line 387, "ResNet18", what does it mean?

Line 390, "and better gets more", please fix the grammar.

References:

Please carefully revise *ALL* the references- remove unnecessary symbols, such as [J], [C],"//", etc. Please add space or "." where it's needed. Please make sure the arXiv numbers are correctly shown.

  1. Response to comment:

Please respond to all the line-by-line comments below.

Line 14, remove "the".

 Response:

Special thanks to you for your comments. We have removed the “the”.

  1. Response to comment:

Line 22, define "P" here, otherwise, it makes no sense to the readers

 Response:

Special thanks to you for your comments. We have defined “P” here. P is defined as the probability that the driver is conservative.

Response to comment:

Line 23, why do exclusively mention LSTM, but not another method that is listed in your tables?

 Response:

Special thanks to you for your comments. We are very sorry for the problem. We have added the VAE+RNN method that is listed in our tables.

  1. Response to comment:

Line 33, ",without" -> ", without".

 Response:

Special thanks to you for your comments. We have corrected the format.

  1. Response to comment:

Line 37, add space between "guaranteed" and "[1]", and the same for line 44's [3], line 47's [4], line 53's [6].

 Response:

Special thanks to you for your comments. We have added space between "guaranteed" and "[1]", and the same for line 44's [3], line 47's [4], line 53's [6].

  1. Response to comment:

Line 44, please fix the pronoun "his".

Response:

Special thanks to you for your comments. We have fixed the pronoun “his” to “its”.

  1. Response to comment:

Line 57, "motion_intentions" -> "motion intentions".

Response:

Special thanks to you for your comments. We have changed “motion_intentions” as “motion intentions”.

  1. Response to comment:

Line 71, "vehicle,which" -> "vehicle, which".

Response:

Special thanks to you for your comments. We have corrected “vehicle,which” as “vehicle, which”.

  1. Response to comment:

Line 89, define RNN.

Response:

Special thanks to you for your comments. We have defined RNN as Recurrent Neural Network.

  1. Response to comment:

Line 100, define CNN.

Response:

Special thanks to you for your comments. We have defined CNN as Convolutional Neural Network.

  1. Response to comment:

Line 106, define C-ResNet-VAE.

Response:

Special thanks to you for your comments. We have defined C-ResNet-VAE as Contrastive ResNet Variational Autoencoder.

  1. Response to comment:

Line 118, define GRU.

Response:

Special thanks to you for your comments. We have defined GRU as Gated Recurrent Unit.

  1. Response to comment:

Line 121, there's only one subsection in this section, you might just rename the section and remove the subsection.

Response:

Special thanks to you for your comments. We are very sorry for the problem of typography. We have renamed the section and removed the subsection.

  1. Response to comment:

Figure 1, please explain what different colors mean.

Response:

Special thanks to you for your comments. We have explained the meaning of the different colors in our paper.

  1. Response to comment:

Lines 130-138, need clarification. Please use the same notations in the text as in Figure 1, change "x", "z" to "X", "Z", or the other way around.

Response:

Special thanks to you for your comments. We are very sorry for the problem of typography. We have corrected “x”, “z” to “X”, “Z”.

  1. Response to comment:

Define "q_{\phi} (z|x)".

Response:

Special thanks to you for your comments. We have defined “q_{\phi} (z|x)” as “The approximate posterior probability distribution of z”.

  1. Response to comment:

Define "P_{\theta}(x|z)" (make it consistent with what's in Figure 1).

Response:

Special thanks to you for your comments. We are very sorry for the problem of typography. We have defined “P_{\theta}(x|z)” as “The approximate posterior probability distribution of x”.

  1. Response to comment:

Define "P_{\theta}(z)".

Response:

Special thanks to you for your comments. We have defined “P_{\theta}(z)” as “hidden variable prior probability distribution”.

  1. Response to comment:

Line 131, "represent" -> "represents".

Response:

Special thanks to you for your comments. We have corrected “represent” as “represents”.

  1. Response to comment:

Line 134-135, please elaborate on low- and high-dimensional.

Response:

Special thanks to you for your comments. We have elaborated on low-and high-dimensional. The high-dimensional represents the observable x-space in the VAE model. The low-dimensional representation decoder reduces the dimensionality of the hidden variable z.

  1. Response to comment:

Line 145, define KL.

Response:

Special thanks to you for your comments. We have defined KL as Kullback-Leibler divergence and added its formula in our paper.

  1. Response to comment:

Equation 1 is incomplete.

Response:

Special thanks to you for your comments. We have completed Equation 1 in our paper.

  1. Response to comment:

Line 152, "obtains" -> "obtain"

Response:

Special thanks to you for your comments. We have corrected “obtains” as “obtain”.

  1. Response to comment:

Line 155, "noise \epsilon", what does it mean?

Response:

Special thanks to you for your comments. We are very sorry for the misrepresentation. We have corrected the mean of noise\epsilon as standard normal distribution.

  1. Response to comment:

Line 188-191, please make sure the "x" and "z" are in line with the text.

Response:

Special thanks to you for your comments. We are very sorry for the problem of typography. We have corrected the “x” and “z” to match the text.

  1. Response to comment:

Line 197, "samples.," -> "samples,".

Response:

Special thanks to you for your comments. We have corrected “samples.,” as “samples,”.

  1. Response to comment:

Line 198, "analyses" -> "analyzes".

Response:

Special thanks to you for your comments. We have corrected “analyses” as “analyzes”.

  1. Response to comment:

Line 206-208, please make sure "G", "x" and "z" are in line with the text, the same for line 212.

Response:

Special thanks to you for your comments. We have corrected the “G”, “x” and “z” to match the text, the same for line 212.

  1. Response to comment:

Equation 2, please define "u", "E", "t", "T_u".

Response:

Special thanks to you for your comments. We have defined “u” as “the set of all driver trajectories”, “E” as “the evidence lower bound”, “t” as “timestep”, “Tu” as “the number of items in the driver trajectories”.

  1. Response to comment:

Line 219, "the more" -> "more".

Response:

Special thanks to you for your comments. We have corrected “the more” as “more”.

  1. Response to comment:

Line 228, "train" -> "training". "insure" -> "ensure".

Response:

Special thanks to you for your comments. We have corrected “train” as “training” and modified “insure” as “ensure”.

  1. Response to comment:

Line 247, define PPO here instead of at line 254.

Response:

Special thanks to you for your comments. We have defined PPO at line 247 in our paper.

  1. Response to comment:

Line 253-255 is a repeat of what has just been said.

Response:

Special thanks to you for your comments. We are very sorry for the problem of typography. We have removed lines 253-255.

  1. Response to comment:

Equation 4, please define every single symbol that has not been defined.

Response:

Special thanks to you for your comments. We have defined every single symbol that has not been defined in our paper.

  1. Response to comment:

Line 265, define IDM.

Response:

Special thanks to you for your comments. We have defined IDM as Intelligent Driver Model.

  1. Response to comment:

Line 267-269, is there a reason why such slow speeds are chosen for the experiments? 2.4 m/s is no faster than a jogger. Also, 0.3 -0.7 m is not a realistic range in real life.

Response:

Special thanks to you for your comments. We set these experimental conditions to be able to better record the number of right turns, timeouts, and collisions completed by self-driving vehicles.

  1. Response to comment:

What is the length of the cars?

Response:

Special thanks to you for your comments. We have set the length of the cars as 5.0 and the width of the cars as 2.0.

  1. Response to comment:

Line 273, define PD.

Response:

Special thanks to you for your comments. We have defined PD as Proportional-Derivative.

  1. Response to comment:

Line 277, define "v_auto".

Response:

Special thanks to you for your comments. We have defined “v_auto” as “the speed of self-driving vehicles”.

  1. Response to comment:

Line 284, please provide more details about the datasets, "generated by Python" provides zero information.

Response:

Special thanks to you for your comments. We have provided more details about the datasets.

  1. Response to comment:

Line 293, define MLP.

Response:

Special thanks to you for your comments. We have defined MLP as Multilayer Perceptron.

  1. Response to comment:

Figure 7 and lines 303-321, please describe in more detail what is shown in the figures.

Response:

Special thanks to you for your comments. We have described in more detail what is shown in the figures.

  1. Response to comment:

What do the lines in them mean?

Response:

Special thanks to you for your comments. We have added the mean of lines in our paper.

  1. Response to comment:

What are the x and y-axis?

Response:

Special thanks to you for your comments. We have explained the x and y-axis in our paper. The x and y-axis is the horizontal and longitudinal displacement in meters, respectively.

  1. Response to comment:

How do you quantify the difference between b, c, d, and e?

Response:

Special thanks to you for your comments. We quantify the difference between b, c, d and e by looking at the degree of separation of the two colors.

  1. Response to comment:

Define "LSTM".

Response:

Special thanks to you for your comments. We have “LSTM” as “Long Short-Term Memory network”.

  1. Response to comment:

Line 333, define "overtime".

Response:

Special thanks to you for your comments. We have defined “overtime” as “The self-driving car failed to make a right turn within the allotted time and did not crash”.

  1. Response to comment:

Table 3, please make sure the whole table is on a single page.

Response:

Special thanks to you for your comments. We are very sorry for the problem of typography. We have ensured the whole table is on a single page.

  1. Response to comment:

Line 358, define "fuzzy trajectories".

Response:

Special thanks to you for your comments. We have defined “fuzzy trajectories” as “very short trajectories and the trajectories with fuzzy front gaps”.

  1. Response to comment:

Line 360, there's no figure 9.

Response:

Special thanks to you for your comments. We are very sorry for the problem of typography. We have corrected figure 9 as figure 8.

  1. Response to comment:

Line 361, the conservative cars are green, not blue.

Response:

Special thanks to you for your comments. We are very sorry for the problem. We have corrected the conservative cars as green.

  1. Response to comment:

Figure 8, please make there is no text in between the figures.

Response:

Special thanks to you for your comments. We have ensured that there is no text between the numbers.

  1. Response to comment:

Please explain what the four small figures mean in (a) and (b), and what the six figures mean in (c).

Response:

Special thanks to you for your comments. We have explained what the four small figures mean in (a) and (b), and what the six figures mean in (c).  

  1. Response to comment:

Line 373, "t+n" what is the size of "n" here?

Response:

Special thanks to you for your comments. We have set the size of n to 50.

  1. Response to comment:

Line 387, "drivers;The" -> "drivers. The".

Response:

Special thanks to you for your comments. We have corrected “drivers;The” as “drivers. The”.

  1. Response to comment:

Line 387, "ResNet18", what does it mean?

Response:

Special thanks to you for your comments. We are using the ResNet18 network architecture. The basic architecture of ResNet18 network is ResNet and the depth of the network is 18 layers. The depth of the network refers to the weight layer of the network, which includes the pooling layer, the activation function, and the linear layer, but not the batching and pooling layer.

  1. Response to comment:

Line 390, "and better gets more", please fix the grammar.

Response:

Special thanks to you for your comments. We are very sorry for the problem of grammar. We have fixed “and better gets more” as “and gets more and more”.

  1. Response to comment:

References:

Please carefully revise *ALL* the references- remove unnecessary symbols, such as [J], [C],"//", etc. Please add space or "." where it's needed. Please make sure the arXiv numbers are correctly shown.

Response:

Special thanks to you for your comments. We are very sorry for the problem of typography. We have corrected the format of reference list according to the Sensors Journal standard.

Thank you very much for all your help and looking forward to hearing from you soon.

Round 2

Reviewer 1 Report

The English of the article should be polished.

Author Response

Very grateful to you to give us sincere advice. We have revised the English of the article and further carefully polished the full text. Thank you very much for your valuable comments.

Reviewer 2 Report

Line 97, "Error! Reference source found" needs to be fixed

Equation 1, there is an empty square in it, meant to be "||"?

Line 279, same problem as Equation 1

Line 281-282, please give a unit to the length and width

Line 296, the link should be moved to the references.

Line 380, fig. -> Fig., same for Line 389

Line 397, n+1 meant t+n?

Line 400, please give "50" a unit

Author Response

Thank you for your letter and for the reviewers’ comments concerning our manuscript entitled “Driver Intent Based Intersection Autonomous Driving Collision Avoidance Reinforcement Learning Algorithm”. Those comments are all valuable and very helpful for revising and improving our paper, as well as the important guiding significance to our researches. We have studied comments carefully and have made correction which we hope meet with approval. The main corrections in the paper and the responds to the reviewer’s comments are as flowing:

Responds to the reviewer’s comments:

Reviewer #2

Comments and Suggestions for Authors

Line 97, "Error! Reference source found" needs to be fixed

Equation 1, there is an empty square in it, meant to be "||"?

Line 279, same problem as Equation 1

Line 281-282, please give a unit to the length and width

Line 296, the link should be moved to the references.

Line 380, fig. -> Fig., same for Line 389

Line 397, n+1 meant t+n?

Line 400, please give "50" a unit

  1. Response to comment:

Comments and Suggestions for Authors

Line 97, "Error! Reference source found" needs to be fixed

 Response:

Special thanks to you for your comments. We have fixed it in the paper.

  1. Response to comment:

Equation 1, there is an empty square in it, meant to be "||"?

 Response:

Special thanks to you for your comments. We have defined “||” here. || is defined as the “or”.

  1. Response to comment:

Line 279, same problem as Equation 1

 Response:

Special thanks to you for your comments. We have defined “||” here. “||” is defined as the “absolute values”.

  1. Response to comment:

Line 281-282, please give a unit to the length and width

 Response:

Special thanks to you for your comments. We have given a unit to the length and width in cm.

  1. Response to comment:

Line 296, the link should be moved to the references.

Response:

Special thanks to you for your comments. We are very sorry for the problem of typography. We have moved the link to the references.

  1. Response to comment:

Line 380, fig. -> Fig., same for Line 389

Response:

Special thanks to you for your comments. We have corrected “fig. -” as “Fig.”.

  1. Response to comment:

Line 397, n+1 meant t+n?

Response:

Special thanks to you for your comments. We are very sorry for the misrepresentation. We have corrected “n+1” as “t+n”.

  1. Response to comment:

Line 400, please give "50" a unit

Response:

Special thanks to you for your comments. We have given “50” a unit as “s”.